Adaptive neural PD controllers for mobile manipulator trajectory tracking

Hernandez-Barragan Jesus josed.hernandezb@academicos.udg.mx
http://orcid.org/0000-0001-7565-0874 D. Rios Jorge
http://orcid.org/0000-0002-9724-1729 Gomez-Avila Javier
Arana-Daniel Nancy
Lopez-Franco Carlos
http://orcid.org/0000-0001-9600-779X Alanis Alma Y.
Department of Computer Science, University of Guadalajara , Guadalajara, Jalisco , México
Liu Pengcheng
Electronic publication date: 2021 Feb 19
Publication date: 2021
Volume: 7
Electronic Location ID: e393
Received 2020 Nov 4; Accepted 2021 Jan 24
Copyright: © 2021 Hernandez-Barragan et al.
Copyright year: 2021
Copyright holder: Hernandez-Barragan et al.
License: This is an open access article distributed under the terms of the Creative Commons Attribution License, which permits unrestricted use, distribution, reproduction and adaptation in any medium and for any purpose provided that it is properly attributed. For attribution, the original author(s), title, publication source (PeerJ Computer Science) and either DOI or URL of the article must be cited.
License URL: https://creativecommons.org/licenses/by/4.0/

Keywords: PID, Adaptive PID, Neural control, Mobile manipulator

Funding: Council of Sciences and Technology (CONACYT), Mexico CB-256769, CB-258068 and PN-2016-4107 This work was supported by Council of Sciences and Technology (CONACYT), Mexico, through the following projects: CB-256769, CB-258068 and PN-2016-4107. The funders had no role in study design, data collection and analysis, decision to publish, or preparation of the manuscript.

==============================
Artificial intelligence techniques have been used in the industry to control complex systems; among these proposals, adaptive Proportional, Integrative, Derivative (PID) controllers are intelligent versions of the most used controller in the industry. This work presents an adaptive neuron PD controller and a multilayer neural PD controller for position tracking of a mobile manipulator. Both controllers are trained by an extended Kalman filter (EKF) algorithm. Neural networks trained with the EKF algorithm show faster learning speeds and convergence times than the training based on backpropagation. The integrative term in PID controllers eliminates the steady-state error, but it provokes oscillations and overshoot. Moreover, the cumulative error in the integral action may produce windup effects such as high settling time, poor performance, and instability. The proposed neural PD controllers adjust their gains dynamically, which eliminates the steady-state error. Then, the integrative term is not required, and oscillations and overshot are highly reduced. Removing the integral part also eliminates the need for anti-windup methodologies to deal with the windup effects. Mobile manipulators are popular due to their mobile capability combined with a dexterous manipulation capability, which gives them the potential for many industrial applications. Applicability of the proposed adaptive neural controllers is presented by simulating experimental results on a KUKA Youbot mobile manipulator, presenting different tests and comparisons with the conventional PID controller and an existing adaptive neuron PID controller.

Introduction

Artificial intelligence (AI) is actively present in our society; AI is used for decades in many relevant areas of our society as the industry, science, entertainment, education, and others (Bryson, 2019). However, it is essential to remark that interest in AI has risen in the last decade. Due to this interest, recently, many works have been reported in the literature in many research areas no name some control, internet of things, natural language processing, machine vision, medicine, robotics, security, social application, among others (Bryson, 2019; Maglogiannis, Iliadis & Pimenidis, 2020).

Proportional Integral, Derivative (PID) controllers are a well-studied kind of controller, which are among the most popular controllers in the industry, mainly for their simplicity (Åström & Hägglund, 1995; Ogata, 2010). The main drawback of PID controllers is that they are only adequate for a nominal process; they have a bad performance under systems uncertainties in operating conditions and changing environmental conditions (Tian, Tadé & Tang, 1999). Even though they are usually the first approach when facing a control problem, even more than other adaptive techniques that have reported more satisfactory results for real-world problems where time delays, unmodeled dynamics and uncertainties are present (Tahoun, 2017b).

It is well-known that there exist techniques to improve the selection of conventional PID parameters; however, most of these techniques are offline methodologies and usually required knowledge about the model of the system, which not always is available (Johnson & Moradi, 2006; Visioli, 2006; Ogata, 2010). The use of artificial intelligence on PID controllers has been used as a tool to improve the performance of PID controllers adapting its parameters online, adjusting them to the changes of the system under consideration. Some of these techniques require access to the complete state of the system and information on its uncertainties and delays, and usually complex calculations (Tahoun 2015, 2017c, 2020). Among these techniques, neural networks stand out; their characteristics allow the implementation of easy, fast, and robust PID controllers known as neural PID controllers, which vary mainly on architecture and training methodology (Rios et al., 2020b; Tahoun & Arafa, 2020; Hernandez-Barragan et al., 2020).

Neural PID controllers learning capabilities allow them to adapt themselves during system operation to unmodeled dynamics, communication time-delays, actuator saturation, among other issues (Ge, Zhang & Lee, 2004; Lopez-Franco et al., 2017; Sarangapani, 2018; Gomez-Avila, 2019), which clearly is a better approach than fixed parameters during the whole operation. Neural adaptive PID controllers have been presented with single neuron and multilayer schemes. The single neuron controllers have three inputs, which are the proportional, derivative, and integral errors. The output of the neuron represents the control action (Rivera-Meja, Léon-Rubio & Arzabala-Contreras, 2012; Jiao et al., 2018; Tang et al., 2020). The multilayer controllers consist of a network with one hidden layer and one node at the output layer. In the hidden layer, three neurons represent the proportional, integral and derivative gains. The neuron of the output layer defines the control action. The inputs of this scheme can be the actual state and the reference (Chen, He & Zhou, 2015; Zeng et al., 2019). But the inputs can also include the proportional, derivative, and integral errors (Sento & Kitjaidure, 2016). Adaptive neural PID controllers trained with the extended Kalman filter (EKF) algorithm based algorithms have proved to show faster learning speed rates and convergence time than adaptive neural PID based on backpropagation training methods, which makes EKF training based neural PID controller more suitable for experimental and real-time tests (Hernandez-Barragan et al., 2020). Also, training algorithms based on Extended Kalman filter (EKF) for neural networks have proven to reliable for recurrent and feedforward neural networks for control applications, presenting real-time applications (Haykin, 2004; Sanchez, Alanis & Loukianov, 2010; Alanis, Arana-Daniel & Lopez-Franco, 2019; Rios et al., 2020a).

Besides the previously mentioned withdraws of PID controllers, a common problem is the windup effect, which is the result of accumulative error action due to the integral part of the controller. This effect produces saturation on actuators and contributes to low-performance, overshoot, high settling time, and instability, losing controllability (Visioli, 2006; Kumar & Negi, 2012; Hernandez-Barragan et al., 2020), which is the reason why anti-windup strategies are important when using PID controllers Tahoun (2017a). Among the proposed anti-windup strategies in the literature are limiter integrator, back-calculation, and observer approach (Visioli, 2006; Kumar & Negi, 2012; Kheirkhahan, 2017; Angel, Viola & Paez, 2019). The integral term is important because it eliminates the steady-state error that the proportional term cannot suppress with a fixed proportional gain. However, the integral action causes oscillations, overshoot, and the windup effect mainly on physical implementation.

Mobile manipulator robots combine mobile platforms and robotic arms, extending operational range and functionality, allowing mobile manipulators to accomplish tasks that are difficult or non-doable for a manipulator or a mobile platform by themselves (Lin & Goldenberg, 2001; Li & Ge, 2017). Among these applications: construction, health-care, nuclear reactor maintenance, manufacturing, military operations, and planetary exploration. Some of those tasks can risk human lives (Lin & Goldenberg, 2001; Li & Ge, 2017). However, such advantages come with complexity and difficulty when designing a control strategy (Li & Ge, 2017). When conventional PID strategies are not enough, and considering what has been previously stated, adaptive intelligent controllers appeared as plausible solutions, especially the ones based on neural networks.

The contributions of this paper are summarized as follows: In Hernandez-Barragan et al. (2020), a single neuron PID controller trained with an extended Kalman filter (EKF) based algorithm and anti-windup effect is proposed, as other neural PID controllers it adjusts itself online during the operation of the system, even with changes in the nature of the problem. However, it requires and anti-wind up methodology. The contribution of this work is the proposal of adaptive neural PD controllers trained with extended Kalman Filter-based algorithms; the adaptive properties of these controllers allow them to compensate for the missing integral part, which can cause the windup effect; in this way, there is no need for an anti-windup methodology as in previous work (Hernandez-Barragan et al., 2020) and at the same time getting a good performance. The proposed EKF-based trained PD controllers, single neuron and multilayer adapt their weights online, eliminating the steady-state error; moreover, oscillations and overshoot are highly suppressed. The performance of the controllers is shown in simulation and experimental results on trajectory tracking tasks for a mobile manipulator robot. In simulations, the proposed controllers are compared against the conventional PID, and an existing single neuron adaptive PID (SNA-PID) controllers (Tang et al., 2020). In real experiments, a study on the robustness of the proposed controller under the presence of disturbances and non-modeled dynamics is presented.

The remaining of this work is organized as follows: first, a summary of the components of PID controllers and adaptive neural PD controllers. Second, the implementation of the proposed neural PD controller on mobile manipulators. Third, simulation and experimental results on a mobile manipulator robot where the performance of proposed controllers is shown; experimental tests are implemented on a KUKA™1 . Finally, conclusions are presented.

Adaptive Neural PD Controllers

A basic PID controller consists of applying the sum of three types of control actions, proportional (P), integral (I) and derivative (D) (Visioli, 2006; Temel, Yagli & Gören, 2013). Moreover, using those control actions, simpler controllers can be obtained, namely, P, PD and PI, which may be enough for some applications, especially linear ones and under regulated conditions. Nevertheless, the PID controller is recognized as the better of them. Even with the existence of more robust control schemes reported in the literature, the popularity of PID is mainly due to its simple implementation. Inspire in this popularity; several works have been proposed to improve the performance of PID controllers. However, most of those works introduce complex methodologies.

Figure 1 Control PID scheme.

P, PI, PD and PID controllers brief summary

P controller

The primary use of the P controller is to reduce the steady-state error of the system. As the proportional gain kP increases, the steady-state error decreases. However, the steady-state error will not be eliminated because increasing kP leads to overshoot, smaller amplitude, phase margin, faster dynamics, and more sensitivity to noise. This control is recommended when the system is tolerable to a constant steady-state error.

PI controller

The use of PI controllers is to eliminate the steady-state error resulting from the P controller. However, it harms the speed of response and system stability. This control is used when the speed of the system is not an issue. PI controller cannot decrease the rise time and eliminate the oscillations, and overshoot is always present.

PD controller

PD controller increases system stability by improving control since it can predict the future error of the system response. Derivative controllers respond to changing error signals, but they do not respond to constant error signals. Due to this, derivative control D is combined with proportional control P.

PID controller

Proportional Integral, Derivative controller needs the derivative gain component in addition to the PI controller to reduce the overshoot and oscillations occurring in the output response of the system. A control scheme of the PID controller is presented in Fig. 1. The manual tuning of the proportional KP, integrative KI and derivative KD gains represent an inconvenience of conventional PID controllers.

Adaptive neuron PD controller

The main disadvantage of conventional PD controllers is that they are not suitable for nonlinear, time-variant systems. adaptive neural controllers are an alternative to overcome this issue.

The proposed adaptive single neuron PD (SNPD) controller is illustrated in Fig. 2. The value e represents the error (1) between the reference yr and the system output y. The inputs x1 and x2 are defined as the proportional (2) and the derivative (3) errors (Moradi, Katebi & Johnson, 2001). The weights ω1 and ω2, are adapted online using the EKF algorithm. The weight ω1 and ω2 represents the proportional gain, and derivative gain, respectively. The value v is computed as the weighted sum of the inputs of the neuron (4). Finally, the output of the neuron y^ is computed with (5), where as activation function is selected tanh(·). The activation function reacts in the range (−1, 1). However, the parameter α can be selected to adjust the control action, since the output of the neuron is directly the control signal u(k)=y^(k).

(1) e(k)=yr(k)−y(k),

(2) x1(k)=e(k),

(3) x2(k)=e(k)−e(k−1),

(4) v(k)=ω1(k)x1(k)+ω2(k)x2(k),

(5) y^(k)=αtanh⁡(v(k)).

The proposed EKF-based training method is described in a section bellow. EKF provides faster learning rates and convergence time than backpropagation, which is crucial for online training.

Figure 2 Adaptive single neuron PID controller.

Adaptive multilayer PD controller

The multilayer network PD (MNPD) scheme is shown in Fig. 3; it consists of a fully connected neural network with one hidden layer with multiples nodes and one node at the output layer. The network input is the error and the derivative between a reference value and the system output. The neural network is trained online using an extended Kalman filter-based algorithm; the objective is to reduce the tracking error by adapting online the output of the network, which is the control signal to the system, it is u(k)=y^(k).

Figure 3 MLP architecture. In this case, the network has one hidden layer whose weights are denoted by and the output layer has one node and its weights are represented with .

Consider a neural network as shown in Fig. 3 with 2 input signals and q nodes in the hidden layer.

The output of the network is given by (6) σi(k)=tanh⁡(ni(k)),i=1…q,

(7) ni(k)=∑j=02ωij(1)(k)xj(k),x0(k)=+1,

(8) v1(k)=∑k=0qω1j(2)(k)uk(k),u0(k)=+1,

(9) y^(k)=v1(k).

Extended Kalman filter based training algorithm for neural networks

For training of neural networks, the weights of the network become the state to be estimated by the EKF, with the objective of reducing the neural network error, which in this case, since the output is the reference minus the system output e(k) = yr(k) − y(k), this is because the neural network output is considered directly as the control signal u(k). The fact that e is minimized means that the neural PD controller output u is working in achieving the control obective of tracking the desired reference yr. The neural network is trained online using an extended Kalman filter-based algorithm (10–12).

(10) K(k)=P(k)H(k)[R(k)+HT(k)P(k)H(k)]−1,

(11) ω(k+1)=ω(k)+ηK(k)e(k),

(12) P(k+1)=P(k)−K(k)HT(k)P(k)+Q(k),

(13) hij(k)=[∂yi(k)∂ωj(k)].

where ω∈Rn is the weight vector, K∈Rn×m is the Kalman gain vector with n as the number of weights, and m the number of outputs of the neural network; P∈Rn×n, Q∈Rn×n, and R∈Rm×m are covariance matrices of weight estimation error, estimation noise, and error noise, respectively; η∈R is the Kalman filter learning rate, and H∈Rn×m is a matrix whose entries hij are the derivative of the neural network output with respect to each weight Eq. (13), yi∈R is the i-th output of the neural network and j = 1 ⋯ n, the error e∈Rm is defined as the difference between the desired output and the neural network output (Sanchez & Alanis, 2006).

Neural network weights are initialized randomly. The Kalman filter learning rate η is selected heuristically to minimized e. It should be considered that if η is sufficiently large, the network could not converge. Conversely, if a lower η is selected, it would take longer to converge. Moreover, matrices P, Q and R are initialized as diagonal matrices with initial values chosen heuristically. The Q matrix is set to deal with process noise, while the R matrix is set to deal with measurement noise. Metaheuristic algorithms can be used to optimize the initial values of the Kalman settings (Villaseñor et al., 2018).

Let us remark that matrices H, K and P are bounded (Song & Grizzle, 1992).

Single neuron EKF training algorithm

The EKF algorithm adjusts onlone the weights ω1 and ω2 for the single neuron. The single neuron scheme is composed with n = 2 (weights) and m = 1 (one output neuron); the dimension of EKF matrices are K∈R2×1, P∈R2×2, Q∈R2×2, R∈R1×1 and H∈R2×1. The weight vector is defined as ω∈R2 that includes ω1 and ω2, and the error e∈R is given by Eq. (1). The matrix H is computed as Eq. (14).

(14) H(k)=[∂y^(k)∂ω1(k)∂y^(k)∂ω2(k)]T=[∂y^(k)∂v(k)∂v(k)∂ω1(k)∂y^(k)∂v(k)∂v(k)∂ω2(k)]T=[αsech2(v(k))x1(k)αsech2(v(k))x2(k)].

Multilayer network EKF training algorithm

The EKF algorithm adjusts online the wights ωij(1)(k) and ωj1(2)(k) for the multilayer network. The multilayer network scheme is set with n (weights) and m = 1 (output neuron). The dimension of EKF matrices are K∈Rn×1, R∈R1×1 and H∈Rn×1. The error e∈R is given by Eq. (1). The matrix H can be expressed as Eq. (16).

(15) H(k)=[∂y^(k)∂w10(1)(k)∂y^(k)∂w11(1)(k)⋯∂y^(k)∂w1q(2)(k)],

(16)  =[γ(n1(k))x0(k)⋯γ(n1(k))xp(k)γ(n2(k))x0(k)⋯γ(nq(k))xp(k)u0(k)u1(k)⋯uq(k)],

with (17) γ(ni(k))=w1i(2)(k)(sech2(ni(k))),i=1,…,q.

Implementation to A Mobile Manipulator for Trajectory Tracking

This section presents a kinematics model for omnidirectional mobile manipulators. Then, the main concepts of differential kinematics are introduced for position control. Finally, the conventional PID and the proposed adaptive PD controllers are provided for the trajectory tracking of omnidirectional mobile manipulators.

Mobile manipulator kinematics

Mobile manipulators are composed of one or more manipulators attached to a mobile platform. Conventional mobile robots such as unicycles, differential drives, and car-like robots increase the workspace of manipulators. However, these platforms have limited movement capabilities due to their nonholonomic kinematics constraints, Li et al. (2016). On the other hand, omnidirectional mobile platforms improved the movement capabilities, allowing moving towards any position and desired orientation (Zhang et al., 2016; Wu et al., 2017; Kundu et al., 2017). This section introduces a kinematic model of a mobile manipulator, which consists of a robotic manipulator of n Degrees of Freedom (DOF) attached to an omnidirectional mobile platform.

The Kinematics chain of mobile manipulators is described in Fig. 4. The homogeneous matrix wTb defines the position and orientation of the mobile platform. The transformation bTm is a constant homogeneous matrix between the mobile platform frame and the manipulator base. The matrix mTe can be computed based on the Denavit-Hartenberg (DH) model of the manipulator (Spong & Vidyasagar, 2008; Lopez-Franco et al., 2018).

Figure 4 Kinematic chain of mobile manipulators.

The transformation wTb is the homogeneous matrix from the world frame w to the mobile platform base frame b, bTm is the homogeneous matrix from b to the manipulator base frame m, mTe is the homogeneous matrix from m to the end-effector frame e.

Considering an omnidirectional mobile platform, the pose of the robot with respect to the world frame w is given by 3 DOF, which are the positions xb and yb, and the orientation θb. Then, the matrix wTb can be defined as Eq. (18).

(18) wTb=[cos⁡(θb)−sin⁡(θb)0xbsin⁡(θb)cos⁡(θb)0yb00100001].

The matrix bTm is constant, and it adjusts the distance from the mobile platform base frame b to the manipulator base frame m. The values tx, ty and tz are used to adjust the distance in the direction of the x-axis, y-axis and z-axis, respectively. If it does not need to adjust the frame orientation, then matrix bTm can be described by Eq. (19).

(19) bTm=[100tx010ty001tz0001].

Let consider a joint variable q to represent the platform configuration qm=[xbybθb]T and the manipulator configuration q<sf>m</sf>=[q1q2q3⋯qn]T, where qi is a joint value for the articulation i. The joint variable for the mobile manipulator is given by q=[qbTqmT]T.

Given the joint variable q, the computation of wTe(q) which is the forward kinematics of the mobile manipulator can be obtained as (20) wTe(q)=wTb(qb)bTmmTe(qm),

where wTe(q) represents the end-effector pose respect to the world frame w. The matrix wTe is expressed as (21) wTe(q)=[r11r12r13txr21r22r23tyr31r32r33tz0001]=[Rt01],

where the orientation of the end-effector is represented by the matrix R, and its Cartesian position is given by the vector t. More information about homogeneous matrices, manipulators kinematics, and forward kinematics can be found in Spong & Vidyasagar (2008), Craig (2005) and Sciavicco & Siciliano (2008).

Differential kinematics

The inverse kinematics consists in the computation of the joint variables q given the end-effector pose 0Tn. This computation can be solved by minimizing an error function using an iterative process based on the differential kinematics (Sciavicco & Siciliano, 2008). Differential kinematics aims to find the relationship between the joint velocities q˙ and the end-effector velocity t˙. The following differential kinematics Eq. (22) gives this relationship (22) t˙=J(q)q˙,

where J is the matrix that relates the contribution of the joint velocities q˙ to the end-effector velocity t˙. The matrix J is called the geometric Jacobian. This Jacobian matrix can be computed as Eq. (23).

(23) J(q)=[∂tx∂q1∂tx∂q2⋯∂tx∂qn∂ty∂q1∂ty∂q2⋯∂ty∂qn∂tz∂q1∂tz∂q2⋯∂tz∂qn],

where t=[txtytz]T is the end-effector position related to the joint variable q=[q1q2⋯qn]T.

An inverse kinematics approach consists in minimizing the error between an actual end-effector position t and the desired position t∗. This error is defined as e=t∗−t. The error e can be mapped to the joint velocities q˙ based on the differential kinematics equation. Eq. (22) is rewritten to compute q˙ given e as Eq. (24).

(24) q˙=J(q)†t˙=J(q)†e,

where J† is the pseudo-inverse of J. The mentioned inverse kinematics approach can be defined as a first-order algorithm that allows the inversion of a motion trajectory, specified at the end-effector position into equivalent joint position and velocities (Sciavicco & Siciliano, 2008).

A robot system with a Jacobian matrix J∈R3×n where n>3, is considered redundant; there are more n DOF than necessary to perform a task with 3 DOF. Commonly, the combination of DOF of the mobile platform and the manipulator represent a redundant robot. In the case of a redundant robot, the solution Eq. (24) can be generalized into Eq. (25).

(25) q˙=J(q)†e+(I−J(q)†J(q))q˙0,

where the first term minimizes the error e, the matrix (I−J†J) allows the protection of vector q˙0 in the null space of J, and I is the identity matrix. In the case that e=0, the result of the second term (I−J†J)q˙0 can reconfigure the joint variable q without changing the end-effector position t.

In this work, it is proposed to design the vector q˙0 to avoid singularities based on the manipulability measure m(q), which is defined as Eq. (26).

(26) m(q)=det(J(q)J(q)T).

Then, vector q˙0 can be computed as Eq. (27).

(27) q˙0=k0(∂m(q)∂q),

where k0 > 0. By maximizing the manipulability measure, redundancy is exploited to move away from singularities. More detailed information about differential kinematics can be found in Spong & Vidyasagar (2008), Craig (2005) and Sciavicco & Siciliano (2008).

PID control design

To solve a position tracking for the mobile manipulator, the controller has to compute the joint velocities q˙(k) at step time k, to control the motion of the mobile manipulator from the actual end-effector position t(k) to the desired position t(k)∗. This section introduces the use of a discrete PID to control the mobile manipulator motion based on the error e(k)=t(k)∗−t(k), which is described as e(k)=[ex(k)ey(k)ez(k)]T.

A discrete PID control (Moradi, Katebi & Johnson, 2001) can be used for each error ex(k), ey(k) and ez(k) as follows (28) ux(k)=KPxex(k)+KIx∑j=1kex(j)+KDx[ex(k)−ex(k−1)],

(29) uy(k)=KPyey(k)+KIy∑j=1key(j)+KDy[ey(k)−ey(k−1)],

(30) uz(k)=KPzez(k)+KIz∑j=1kez(j)+KDz[ez(k)−ez(k−1)],

where KPx, KIx and KDx are the proportional, integrative and derivative gains for error ex, respectively. Similarly, the parameters KPy, KIy and KDy are the gains for error ey, and KPz, KIz and KDz are the gains for error ez. The control output u(k)=[ux(k)uy(k)uz(k)]T can be mapped to the joint velocities q˙(k) based on Eq. (25) to control the system. This is (31) q˙(k)=J(q(k))†u(k)+(I−J(q(k))†J(q(k)))q˙0.

Neural PD controllers implementation

Both proposed neural PD presented in previous sections are implemented on the above-described mobile manipulator. Figure 5 shows the general control scheme for both implementations.

Figure 5 Adaptive neural PD control scheme for the position control of mobile manipulators.

The block called Adaptive Neural, can represent the single neuron scheme or the multilayer network scheme.

An adaptive neural PD control module is designed to minimize the error ex, ey and θz. Each control signal (neural PD output) ux, uy and uz, are compute for each control module. These control signals u(k)=[ux(k)uy(k)uz(k)]T are mapped to the joint velocities q˙(k) using Eq. (31) to control the system.

If expression Eq. (31) is multiplied by the Jacobian matrix J(q(k)), then we have (32) J(q(k))[q˙(k)]=J(q(k))[J(q(k))†u(k)+(I−J(q(k))†J(q(k)))q˙0],

(33) =J(q(k))J(q(k))†u(k)+J(q(k))(I−J(q(k))†J(q(k)))q˙0,

(34) =u(k)+(J(q(k))−J(q(k))J(q(k))†J(q(k)))q˙0,

(35) =u(k)+(J(q(k))−J(q(k)))q˙0,

(36) =u(k),

which indicates that u(k)=J(q(k))q˙(k) is a solution of the differential kinematics of the system, where the matrix J(q(k)) is bounded (Hernandez, Nuño & Alanis, 2016). Moreover, the computed control signal of the neural controller Eq. (5) is also bounded. Considering that neural control for system Eq. (31) is a feed forward network control law, it composed a stable system (Khalil, 2002).

Results

This section shows through simulation and experimental test the performance of the proposed controllers, the adaptive single neuron PD (SNPD) and multilayer network PD (MNPD). The controllers are compared against conventional PID controller, and an existing single neuron adaptive PID (SNA-PID) (Tang et al., 2020). Reference trajectories are selected with different degrees of difficulty for both simulations, and experimental tests on the KUKA Youbot™ mobile manipulator, see Fig. 6. Moreover, experiments show how the controllers behave in the presence of disturbances and non-modeled dynamics.

Figure 6 Omnidirectional mobile manipulator KUKA Youbot™.

Photo credit: Jesus Hernandez-Barragan.

The KUKA Youbot™ is composed of a manipulator of 5 DOF, and an omnidirectional mobile platform of 3 DOF. Respect to the mobile manipulator kinematics, the transformation wTb can be computed with the mobile platform pose, which is given by xb, yb and θb, see Eq. (18). The constant transformation bTm is considered to be Eq. (37).

(37) bTm=[1000.14001000010.1510001].

The values shown in Eq. (37) were obtained based on the KUKA Youbot™ technical specifications. Finally, the DH table in Table 1, is used to compute the transformation mTe. The joint variable q for the mobile manipulator is Eq. (38).

(38) q=[xbybθbθ1θ2θ3θ4θ5]T,

where the joint values θ1−θ5 represent the joint configuration of the manipulator.

Table 1 DH table for KUKA Youbot™ manipulator.

Values a, α and d are parameters of the DH convention.

Joint	a (mm)	α (rad)	d (mm)	θ (rad)	
1	33	π/2	147	θ1	
2	155	0	0	θ2	
3	135	0	0	θ3	
4	0	π/2	0	θ4	
5	0	0	217.5	θ5	

For simulations and experimental test, the weights for both SNPD and MNPD controllers are set randomly in every trajectory test. The parameter setting for the EKF are: matrices P and Q are initialized as diagonal matrices with Pii=1 and Qii=0.1 with i = 1,2,⋯,n, the parameter R=0.001, the Kalman filter learning rate η = 0.2 and α = 1. These parameters were chosen heuristically. For PID controller, proportional gains are set as KPx=KPy=KPz=1.5, integrative gains KIx=KIy=KIz=0.001, and derivative gains KDx=KDy=KDz=0.5. The gains of the PID controller are also selected heuristically. The weights for the SNA-PID controller are set randomly. Additionally, the SNA-PID learning rates are tuned to ηx = ηy = ηz = 1.0 × 10−5 and the proportional coefficient is set to K = 2.5. The setting for SNA-PID is tuned based on Tang et al. (2020).

The considered trajectories, at step time k are generated as follows: Circulartrajectoryxr(k)=0.5,yr(k)=0.05cos⁡(0.2kπ),zr(k)=0.45+0.05sin⁡(0.2kπ).

Rosecurvetrajectoryxr(k)=0.5,yr(k)=r(k)cos⁡(0.2kπ),zr(k)=0.45+r(k)sin⁡(0.2kπ),r(k)=0.035+0.015cos⁡(0.6kπ).

Trapezoidaltrajectoryxr(k)=0.5,yr(k)=0.1∗k,r(k)=0.45+0.08sin⁡(2yr(k)π),zr(k)={0.5ifr(k)>0.50.4ifr(k)<0.4r(k)otherwise.

Sinusoidaltrajectoryyr(k)=0.1∗k,xr(k)=0.5+0.05cos⁡(2yr(k)π),zr(k)=0.45+0.05sin⁡(2yr(k)π).

The desired position for the end-effector is defined as t→(k)∗=[xr(k)yr(k)zr(k)]T. The circular and rose curve trajectories are considered for simulations. The rose curve, trapezoidal and sinusoidal trajectories are considered for real experiments.

Simulations

The first trajectory for simulation is the circular. Although conventional PID controller presents a good response, their gains remain constant, and they cannot adapt to changes in the system operating conditions. On the other hand, the MNPD, SNA-PID, and SNPD approaches can correctly follow the reference once the weights are adjusted.

The system response results for the circular trajectory are given in Fig. 7. The settling time is almost the same for all the approaches. The conventional PID presents overshoot and steady-state errors. The SNA-PID controller suppresses overshoot, but it has small steady-state errors. Although MNPD presents oscillations while the weights are adapting, the neural algorithms can follow the sinusoidal trajectory better than the PID and SNA-PID. Moreover, the SNPD controller does not suffer from overshoot.

Figure 7 System response for circular trajectory.

(A–C) System responses for the x-axis, y-axis, and z-axis, respectively.

The root mean square (RMS) and median absolute deviation (MAD) for the circular trajectory are presented in Table 2. The adaptive approaches present the best results, which are highlighted in bold. In this case, the SNPD control scheme reportesd the smallest RMS results in general.

Table 2 Simulation results for the circular trajectory.

The best results are highlighted in bold.

Measure	Method	ex	ey	ez	
RMS	MNPD	8.6035 × 10−4	2.5297 × 10−3	6.7063 × 10−3	
	SNA-PID	8.1755 × 10−4	9.0910 × 10−3	9.9509 × 10−3	
	PID	1.0547 × 10−3	1.2872 × 10−2	1.3803 × 10−2	
	SNPD	7.8284 × 10−4	2.1269 × 10−3	3.5693 × 10−3	
MAD	MNPD	1.3391 × 10 −4	5.5760 × 10−4	1.3227 × 10−3	
	SNA-PID	1.8123 × 10−4	8.0183 × 10−3	8.0622 × 10−3	
	PID	2.1417 × 10−4	1.1419 × 10−2	1.1686 × 10−2	
	SNPD	1.2753 × 10−4	6.0505 × 10−4	6.4863 × 10−4	

In Fig. 8, the trajectory tracking and velocity control signals for the circular trajectory are presented. From Figs. 8A to 8D, the PID control passes over the reference caused by the integral part, and MNPD presents oscillations as expected from the system response. However, MNPD, SNA-PID, and SNPD controllers follow the trajectory correctly. As seen from Figs 8E to 8H, at the first steps adaptive weights compute bigger control signals than PID. However, it is necessary to reach the reference with a small tracking error.The adaptation ability of both MNPD and SNPD is also shown.

Figure 8 Trajectory following and velocity control signal results for the circular trajectory.

(A–D) The trajectory following results. (E–H) The velocity control signal results.

The same gains and parameters for the four approaches are set for a new desired trajectory to be tested, and the results are shown in Fig. 9. Similar results can be seen in the system response. The settling time is similar, and the MNPD present oscillations during the adaptations of its weights. In this case, PID and SNA-PID controllers present steady-state errors. However, SNA-PID performs better than PID. Moreover, The PID controller also has overshoot. The best results are given by SNPD and MNPD.

Figure 9 System response results for the rose curve trajectory.

(A–C) System responses for the x-axis, y-axis, and z-axis, respectively.

Table 3 shown the RMS and MAD results for the rose curve trajectory. The adaptive scheme has been demonstrated to have better results than the conventional PID controller. In this case, MNPD controller shows the smallest RMS results in general. In contrast, SNPD shows the best MAD results.

Table 3 Simulation results for the rose curve trajectory.

The best results are highlighted in bold.

Measure	Method	ex	ey	ez	
RMS	MNPD	6.9811 × 10−4	2.1346 × 10−3	4.4524 × 10−3	
	SNA-PID	8.1755 × 10−4	7.9247 × 10−3	9.0976 × 10−3	
	PID	1.0547 × 10−3	1.1191 × 10−2	1.2804 × 10−2	
	SNPD	7.7519 × 10−4	2.1415 × 10−3	3.5652 × 10−3	
MAD	MNPD	8.7982 × 10−5	3.6619 × 10−4	4.9729 × 10−4	
	SNA-PID	1.8123 × 10−4	6.8568 × 10−3	7.1826 × 10−3	
	PID	2.1404 × 10−4	9.1301 × 10−3	9.9235 × 10−3	
	SNPD	1.2586 × 10−4	5.8940 × 10−4	6.4607 × 10−4	

In Fig. 10, the trajectory following, and the velocity control signals for the rose curve trajectory are reported. From Figs. 8A to 8D can be noticed that the adapting approaches outperform the conventional PID controller. The SNA-PID controller shows a small steady-state error, while MNPD and SNPD report the smallest. The SNA-PID control improved the performance of PID, but it is needed to tune the proportional coefficient K to improve the performance. As can be seen from Figs. 8E to 8H, at the beginning of the trajectory, the weights adaptation of the MNPD and the SNPD compute bigger control signals than SNA-PID and PID; however, this is necessary to reach the reference with a small tracking error.

Figure 10 Trajectory following and velocity control signal results for the rose curve trajectory.

(A–D) The trajectory following results. (E–H) The velocity control signal results.

Experiments

Real experiments are carried out in two parts. In the first part, the proposed SNPD controller is compared against the classical PID. In this case, two trajectories are tested for comparison purposes. In the second part, the robustness of the SNPD controller is tested under the presence of disturbances and non-modeled dynamics.

In the first part of the experiments, the adaptive SNPD and MNPD controllers performed similarly in simulations. However, MNPD shows oscillations during the adaptations of its weights at the beginning. These oscillations can be eliminated if pre-trained weights are used instead of initializing them randomly every time. Thus, it is considered to compare the SNPD controller to the PID controller since PID performed better than PD. Moreover, the same gains and parameters used for simulation were used for real-time experiments. The weights in the SNPD were randomly initialized.

In Fig. 11, the system response for both approaches is shown. The real system is not the same in simulation, and the gains of the conventional PID must be tuned again. Otherwise, it will not be able to follow the trajectory correctly and present a longer settling time. In contrast, using the same parameters as in simulation, the SNPD was able to adapt and showed a shorter settling time.

Figure 11 System response results for the rose curve trajectory in real experiments.

(A–C) System responses for the x-axis, y-axis, and z-axis, respectively.

Table 4 reports the RMS and MAD results for the rose curve trajectory in real experiments. The SNPD scheme has been demonstrated to have better results than conventional PID with the smallest RMS and MAD results in general.

Table 4 Experimental results for the rose curve trajectory.

The best results are highlighted in bold.

Measure	Method	ex	ey	ez	
RMS	SNPD	3.7452 × 10 −3	3.3248 × 10 −3	8.4861 × 10 −3	
	PID	2.9319 × 10 −3	1.7032 × 10 −2	2.1101 × 10 −2	
MAD	SNPD	9.5960 × 10 −4	1.1829 × 10 −3	2.0750 × 10 −3	
	PID	1.3942 × 10 −3	1.2963 × 10 −2	1.6109 × 10 −2	

The trajectory following and velocity control signals for the rose curve trajectory are illustrated in Fig. 12. In Figs. 12A and 12B, the response for the rose curve trajectory is shown. As can be seen, PID cannot follow the trajectory correctly, and it is confirmed in Table 4. In Figs. 12C and 12D, adaptive SNPD computes bigger control signals than PID. However, this demonstrates that SNPD is adjusting itself to reject perturbation and changes during experimental tests.

Figure 12 Trajectory following and velocity control signal results for the rose curve trajectory in real experiments.

(A–B) The trajectory following results. (C–D) The velocity control signal results.

A new trajectory is tested, and the system response results are shown in Fig. 13. The SNPD control performed better than PID for the results for the trapezoidal trajectory. The results exhibited the adaptation ability of the SNPD, while PID control requires the tune of its gains.

Figure 13 System response results for the trapezoidal trajectory in real experiments.

(A–C) System responses for the x-axis, y-axis, and z-axis, respectively.

The trajectory following and velocity control signals results for the trapezoidal trajectory are given in Fig. 14. In Figs. 14A and 14B, the PID scheme reported bigger tracking error than SNPD controller. From Figs. 14C and 14D, it is clear that bigger control action is required to be able to follow the trajectory with minimum error tracking. This is achieved with the online adaptation of SNPD controller.

Figure 14 Trajectory following and velocity control signal results for the trapezoidal trajectory in real experiments.

(A–B) The trajectory following results. (C–D) The velocity control signal results.

Finally, Table 5 reported the RMS and MAD results for the trapezoidal trajectory in real experiments. The SNPD scheme outperformed the PID controller with the smallest RMS and MAD results in general.

Table 5 Experimental results for the trapezoidal trajectory.

The best results are highlighted in bold.

Measure	Method	ex	ey	ez	
RMS	SNPD	3.8064 × 10 −3	3.4122 × 10 −3	8.1596 × 10 −3	
	PID	3.0025 × 10 −3	5.5684 × 10 −2	1.8825 × 10 −2	
MAD	SNPD	8.8564 × 10 −4	1.3090 × 10 −3	2.0708 × 10 −3	
	PID	1.7514 × 10 −3	3.5206 × 10 −2	1.5032 × 10 −2	

In the second part of the experiments, three tests were performed. In Test 1, a trajectory tracking on a smooth floor is considered. This test represents an ideal environment. In Test 2, the same trajectory tracking is performed on a rough floor. This test represents the presence of disturbances. Finally, in Test 3, the same trajectory tracking is performed on the uneven floor with a load on board of 1.5 kg. This last test represents non-modeled dynamics. Moreover, since the experiments were carried out on a real robot, there was measurement noise.

The SNPD controller is considered for this test under the sinusoidal trajectory. The results for the system response are provided in Fig. 15. The system response results are very similar for the three tests. This indicates that no adjustment to the controller parameters is required to handle uncertainties in unmodeled changes in system dynamics. Moreover, there is no presence of steady-state errors, and the controller reaches the references within an adequate settling time.

Figure 15 System response results for the sinusoidal trajectory in real experiments.

(A–C) System responses for the x-axis, y-axis, and z-axis, respectively.

The trajectory following and velocity control signals for these tests are illustrated in Fig. 16. In Fig. 16A, the results show the correct trajectory following as expected from the system response results. As can be seen in Figs. 16C and 16D, the SNPD computes bigger control signals than the results in Fig. 16B. This shows that the weights are dynamically adapted to reject disturbances and changes in dynamics during the experimental test

Figure 16 Trajectory following and velocity control signal results for the sinusoidal trajectory in real experiments.

(A) The trajectory following results. (B–D) The velocity control signal results.

Finally, the Table 6 shows the RMS and MAD results for the sinusoidal trajectory. The SNPD scheme has been demonstrated to have better results than conventional PID with the smallest RMS and MAD results in general. There is not a big difference between the RMS and MAD results, which indicates that the proposed SNPD controller is robust to unmodeled dynamics and disturbances.

Table 6 Experimental results for the sinusoidal trajectory in real experiments.

Measure	Test	ex	ey	ez	
RMS	Test 1	7.7389 × 10 −3	5.0422 × 10 −3	1.2476 × 10 −2	
	Test 2	8.2315 × 10 −3	7.2873 × 10 −3	1.2537 × 10 −2	
	Test 3	8.2503 × 10 −3	6.8967 × 10 −3	1.2678 × 10 −2	
MAD	Test 1	2.4092 × 10 −3	1.9660 × 10 −3	4.1409 × 10 −3	
	Test 2	2.9295 × 10 −3	2.5150 × 10 −3	4.1397 × 10 −3	
	Test 3	2.8684 × 10 −3	4.1865 × 10 −3	4.1863 × 10 −3	

Conclusions

In this work, an adaptive single neuron PD (SNPD) and multilayer network PD (MNPD) controllers trained with the EKF algorithm were proposed. The performance of these approaches was considered for trajectory tracking of the KUKA Youbot mobile manipulator. Simulation and real experiments were performed to compare the classical PID controller against the proposals. An existing adaptive single neuron controller (SNA-PID) is also considered for comparison. Simulation and experiment results reported that conventional PID control reported overshoot and steady-state errors. The SNA-PID controller highly reduced the overshoot, but it presented small steady-state errors. In contrast, the adaptive neural PD controllers eliminated the steady-state error and highly suppressed the overshoot in general. Moreover, adaptive PD schemes show better settling time and high performance with smaller tracking results. The results also showed that even without an integral part, the PD neural controllers trained with extended Kalman filter offer better overall performance than a conventional PID. They present a small overshoot, and the offset is reduced. Additionally, the experimental results indicate that the SNPD controller shows a superior system response under perturbations and changes during the operation that the PID controller. The conventional PID controller requires the tuning of its gains to improve the performance. The SNPD controller shows better performance than MNPD, mainly due to more weights present in MNPD. It is shown that they present similar settling times, and the oscillations present with MNPD can be eliminated if trained weights are used instead of initializing them randomly every time. However, it was exposed that this is unnecessary, and both approaches exhibit good adaptation to uncertainties in the system. One of the main reasons for PI, PD, and PID controllers’ success is their implementation simplicity. Some works have been proposed to deal with the drawbacks of the conventional PID, adding in some cases a fair complexity at implementation time. The proposed adaptive neural PD controllers are easy to implement, having good performances. Finally, the presented approach considers the system kinematics to compute a motion trajectory, specified at the end-effector position into equivalent joint position and velocities. As future work, the system dynamics can be considered to compute a motion trajectory in terms of position, velocities, and acceleration based on second-order algorithms.

Additional Information and Declarations

Competing Interests

Author Contributions

Data Availability

1 KUKA is a registered trademark of KUKA AG.

Alma Yolanda Alanis Garcia is an Academic Editor for PeerJ.

Jesus Hernandez-Barragan conceived and designed the experiments, performed the experiments, performed the computation work, prepared figures and/or tables, authored or reviewed drafts of the paper, and approved the final draft.

Jorge D. Rios conceived and designed the experiments, performed the experiments, prepared figures and/or tables, authored or reviewed drafts of the paper, and approved the final draft.

Javier Gomez-Avila conceived and designed the experiments, performed the experiments, performed the computation work, prepared figures and/or tables, authored or reviewed drafts of the paper, and approved the final draft.

Nancy Arana-Daniel analyzed the data, authored or reviewed drafts of the paper, supervision and project administration, and approved the final draft.

Carlos Lopez-Franco analyzed the data, authored or reviewed drafts of the paper, supervision and project administration, and approved the final draft.

Alma Y. Alanis analyzed the data, authored or reviewed drafts of the paper, supervision and project administration, and approved the final draft.

The following information was supplied regarding data availability:

Raw data and code are available at GitHub:

https://github.com/JesusHernandezBarragan/Adaptive-neural-PD-controllers-for-mobile-manipulator-trajectory-tracking.git.

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
