# Peer review of "Adaptive neural PD controllers for mobile manipulator trajectory tracking"

_PeerJ Computer Science, doi:10.7717/peerj-cs.393_

## Round 0.1 · original submission · Major Revisions

This manuscript studies an interesting topic and the authors have presented some interesting results. However, two reviewers have pointed out some issues to be considered and addressed, e.g., comparing the experimental results with some prevailing methods, analysing the robustness of the controller, improving English writing, correcting the typos and grammatical errors, etc. Please carefully revise the manuscript based on the comments and suggestions.

Reviewer 1 ·

Basic reporting

no comment

Experimental design

no comment

Validity of the findings

no comment

Additional comments

My main comments are summarized as follows:

1- The PID control is well studied in the literature, the main contribution of your work and the main differences between the existing work should be declared in the introduction.
2- The literature review is not enough.
3- In the industry, all control systems work in noisy environment in which there exist faults, disturbances/noise and delays, what about these uncertainties in your work. Please discuss
• Fault-tolerant control for a class of quantised networked control of nonlinear systems with unknown time-varying sensor faults, International Journal of Control 93 (3), 619-628.
• A new online delay estimation-based robust adaptive stabilizer for multi-input neutral systems with unknown actuator nonlinearities, ISA transactions 2017, 139-148.
• Less-conservative robust adaptive control of neutral systems with mixed time-delays, International Journal of Systems Science 48 (4), 675-685.
• Anti-windup adaptive PID control design for a class of uncertain chaotic systems with input saturation, ISA transactions 66, 176-184.
• Adaptive stabilization of neutral systems with nonlinear perturbations and mixed time‐varying delays, International Journal of Adaptive Control and Signal Processing 29, 1328-1340.
• A new unmatched-disturbances compensation and fault-tolerant control for partially known nonlinear singular systems, ISA Transactions 104, September 2020, 310-320
• ….
4- In Fig. 2, the author used derivative operator d/dt although they deal with difference equations. Please correct it.
5- You should compare your experimental results with the existing results.
6- Some remarks should be added to declare the effects of the design parameters in the simulation results.

Reviewer 2 ·

Basic reporting

This paper presents adaptive neural PD controllers applied to mobile manipulator trajectory tracking. The paper uses clear, unambiguous, technically correct text. and its structure is consistent and easy to follow. Raw data is shared and supports the results presented.

Experimental design

The paper is into the scope of the journal. It is suggested that the methods should be described with sufficient detail to be reproducible by another investigator.

Validity of the findings

The proposed controllers are interesting, however an important aspect is related to stability. This study must be included within the paper. Furthermore, the proposed control only takes into account the kinematics of the robot. It would be important that the dynamics of the robot be taken into account. Finally, it is important to analyze the robustness of the controllers in the presence of disturbances, non-modeled dynamics and measurement noise.

---

## Round 0.2 · accepted · Accept

The manuscript has been improved based on the reviewers' comments and suggestions.

Reviewer 1 ·

Basic reporting

accept

Experimental design

accept

Validity of the findings

accept

Additional comments

My recommendation is "acceptance"

Reviewer 2 ·

Basic reporting

.

Experimental design

.

Validity of the findings

.

Additional comments

The reviewer thanks the authors for the response given to each of the concerns raised in the review process. The reviewer considers that this manuscript can be considered for possible publication.